# Exploring the Phytochemicals and Anti-Cancer Potential of the Members of Fabaceae Family: A Comprehensive Review

**DOI:** 10.3390/molecules27123863

**Published:** 2022-06-16

**Authors:** Muhammad Usman, Waseem Razzaq Khan, Nousheen Yousaf, Seemab Akram, Ghulam Murtaza, Kamziah Abdul Kudus, Allah Ditta, Zamri Rosli, Muhammad Nawaz Rajpar, Mohd Nazre

**Affiliations:** 1Department of Botany, Government College University Lahore, Katchery Road, Lahore 54000, Pakistan; usmanphytologist@gmail.com (M.U.); dr.nousheenyousaf@gcu.edu.pk (N.Y.); 2Institut Ekosains Borneo, Universiti Putra Malaysia Kampus Bintulu, Bintulu 97008, Malaysia; khanwaseem@upm.edu.my; 3Department of Biology, Faculty of Science, Universiti Putra Malaysia, Serdang 43400, Malaysia; seemabakram@ymail.com; 4Faculty of Environmental Science and Engineering, Kunming University of Science and Technology, Kunming 650500, China; murtazabotanist@gmail.com; 5Department of Forestry Science and Biodiversity, Faculty of Forestry and Environment, Universiti Putra Malaysia, Serdang 43400, Malaysia; kamziah@upm.edu.my; 6Department of Environmental Sciences, Shaheed Benazir Bhutto University Sheringal, Upper Dir 18000, Pakistan; 7School of Biological Sciences, The University of Western Australia, 35 Stirling Highway, Perth, WA 6009, Australia; 8Department of Forestry Science, Faculty of Agriculture and Forestry Sciences, Universiti Putra Malaysia Kampus Bintulu, Bintulu 97008, Malaysia; zamrirosli@upm.edu.my; 9Department of Forestry, Faculty of Life Sciences, SBBU Sheringal, Dir Upper 18000, Pakistan; rajparnawaz@gmail.com

**Keywords:** cancer, Fabaceae, phytochemicals, cancer treatment, anti-oxidants, apoptosis

## Abstract

Cancer is the second-ranked disease and a cause of death for millions of people around the world despite many kinds of available treatments. Phytochemicals are considered a vital source of cancer-inhibiting drugs and utilize specific mechanisms including carcinogen inactivation, the induction of cell cycle arrest, anti-oxidant stress, apoptosis, and regulation of the immune system. Family Fabaceae is the second most diverse family in the plant kingdom, and species of the family are widely distributed across the world. The species of the Fabaceae family are rich in phytochemicals (flavonoids, lectins, saponins, alkaloids, carotenoids, and phenolic acids), which exhibit a variety of health benefits, especially anti-cancer properties; therefore, exploration of the phytochemicals present in various members of this family is crucial. These phytochemicals of the Fabaceae family have not been explored in a better way yet; therefore, this review is an effort to summarize all the possible information related to the phytochemical status of the Fabaceae family and their anti-cancer properties. Moreover, various research gaps have been identified with directions for future research.

## 1. Introduction

Cancer is a very dangerous disease and is characterized by the uncontrollable growth of cells. The number of cases and deaths due to cancer is increasing with every passing day (Figure 1). It is difficult to control and has become a major concern for scientists around the world [1]. Many stages (initiation, promotion, and progression) occur in the formation of the cancerous cells [2]. Irregular rates of dietary imbalance, hormonal imbalance, chronic infections, inflammation, and smoking are the major causes of cancer [3]. Despite several treatments to cure cancer, it is still considered the second most devastating cause of death around the world [4,5]. Various methods have been employed to treat cancer, e.g., stem cell transplantation, chemotherapy, radiotherapy, immunotherapy, and surgery. The most effective method to treat cancer is chemotherapy, but various side effects are associated with this method [1,6]. Due to the various side effects of radiotherapy and chemotherapy, alternative treatment methods with no or few side effects are required for the prevention and treatment of cancer [7]. Recently, researchers around the globe have focused their efforts on discovering novel drugs from natural sources such as plants with authentic medicinal importance [8].

Herbal treatment is a natural gift for humans to use to improve their health [9]. Since ancient times, plants and their phytoconstituents have been used as far as medicinal purposes are concerned [10]. Podophyllotoxin was discovered in the 1960s at the same time as when cancer treatments were being searched for from therapeutic plants, which contributed to the discoveries of taxol, vinblastine, camptothecin, and vincristine [11,12,13]. Many plants and their phytochemicals have the potential to control the spread of cancer in the body and continue to attract researchers to examine the anti-cancer activities of various extracted phytochemicals from plant sources [14].

**Figure 1 molecules-27-03863-f001:**
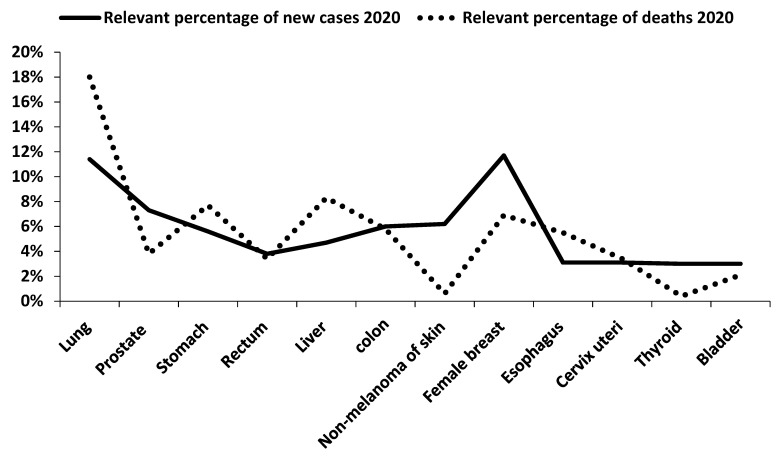
Relevant percentage of new cases and deaths caused by some major types of cancers in 2020 (Sung et al. [15]).

Kingdom Plantae is characterized by approximately 250,000 plant species; however, the real issue is that only 10% of all plant species have been tested for the treatment of cancer [11,12]. Anti-cancer compounds occur in various plant parts, e.g., leaf, flower, fruits, roots, stigmas, pericarp, embryo, rhizomes, seeds, stem, sprouts, and bark, and these phytochemicals are famous for their role in pharmacology [2]. Different phytochemicals such as flavonoids, alkaloids, saponins, terpenes, lignin, vitamins, minerals, taxanes, gums, biomolecules, glycosides, oils, and various other metabolites are known to show anti-cancer activities [2]. These compounds play a vital role in cancer prevention by activating enzymes and proteins, regulating cellular and signaling events in growth, their anti-inflammatory effect, and anti-oxidant action [12,16].

The Fabaceae family has a diverse fossil record where the late Paleocene period represents the oldest fossil records of the family, which shows the history of the Fabaceae family [17]. Most researchers believe that members of this family have evolved in arid and semiarid regions near the Tethys Sea [18]. These plants have been the main part of meals in these regions since 6000 BC because of their richness in proteins. There is also a history of the use of these plants by humans in Asia, Europe, and North America for medicinal purposes [18,19]. Currently, species of the Fabaceae family occur naturally or are cultivated everywhere around the globe except the poles [18,20]. Different types of beans are used in cuisines due to their richness of proteins and health-promoting activities in the Middle East, Asia, Mexico, and South America [20].

Phytochemicals of this family have industrial and pharmacological importance [21,22]. This family is a big source of phytochemicals, namely, flavonoids, lectins, saponins, alkaloids, carotenoids, and phenolic acids, which have an anti-cancer property, and the use of these phytochemicals is increasing over time [13,23]. However, phytochemicals of this family have not been explored massively for their effect on cancer cell growth. Therefore, more research is needed in the future to explore the potential of phytochemicals of the Fabaceae family against cancer and to discover novel drugs against this disease. Various researchers have worked on anti-cancer aspects of the medicinal plants from the Fabaceae family. This review is the first attempt to explore the potential effects of phytochemicals of the Fabaceae family against cancer cell growth, development, and associated mechanisms. Moreover, research gaps have been explored and future recommendations are given.

## 2. Development of Cancer and Phytochemical Pathways of Action

Various researches have been conducted over the passing time to understand the exact process of carcinogenesis. Sporn and Liby [24] demonstrated that carcinogenesis is a multistep process, which is divided into three main phases, i.e., initiation, promotion, and finally progression. In most instances, a carcinogen is detoxified within the body as it enters. However, it may be activated through various metabolic pathways. According to Klaunig and Wang [25], carcinogenic agents increase oxidative stress and damage the DNA, and lead to the initiation of carcinogenesis. The proliferation activity of initiated cells starts during the promotion phase and leads to the preneoplastic cells’ accumulation. These preneoplastic cells begin invading and spreading in different parts of the body during the third and the last phase, i.e., the progression phase. The progression phase is irreversible as seen in Figure 2 [26].

The prevention and treatment of cancer by only one pathway does not turn out to be an effective way due to the involvement of multiple pathways in the occurrence as well as the progression of cancer [27,28]. All the treatment strategies are subjected to a few hindrances, e.g., side effects and drug resistance to chemotherapy [29]. These hindrances have made it difficult for scientists to efficiently develop various treatment strategies related to cancer [30,31]. Chemoprevention is another approach that is widely practiced worldwide, and it is useful during the initiation phase of carcinogenesis, while some have even reported its effectiveness in the promotion and progression phases too [32]. The chemopreventive agents are generally classified into two principal categories, where one group includes blocking agents, while others are suppressive agents majorly sourced from plant phytochemicals [33]. Blocking agents work uniquely; they suppress the carcinogen activation through the metabolic pathway and do not allow carcinogenic agents to interact with the biomolecule. On the other hand, suppressive agents work in another way and suppress the promotion or progression of cancerous cells [34]. The chemopreventive agents usually have an anti-proliferative and anti-oxidant effect or regulate specific enzyme activities and cell cycles. Furthermore, these agents also regulate signal transduction pathways and prevent carcinogenesis [35]. The phytochemicals’ pathway of the anti-cancer effect is presented in Figure 3.

## 3. Steps Involved in the Development of Phytochemical Drugs from the Medicinal Plants

The quality of active phytochemicals in plants determines their ability to be used as therapeutic agents. Other vital factors, which affect the quality of phytochemicals in plants, are the age of the plants, climate, and season. On the other hand, some plant parts have higher levels of bioactive phytochemicals than others, but more research is needed to improve the knowledge of phytochemicals and how these phytochemicals could be exploited for cancer prevention and treatment (Figure 4). Many techniques can be used to purify the active phytochemical including isolation assays, combinatorial chemistry, and bioassay-guided fractionation [13].

Several analytical techniques can be used for the separation of bioactive compounds from a mixture of compounds in the case of bioassay-guided fractionation. Natural extract tests from the dry or wet plant material serve as the beginning process to evaluate biological activity [13]. For the fractionation of active extract, suitable matrices are utilized, and various analytical techniques, namely, mass spectroscopy, HPLC, TLC, FTIR, and NMR, are used for the separation of active compounds. There is a great variety of solvents that can be used for the separation. For the fractionation, silica, superdex, and other suited matrices can be used. Various dyes can be used to detect the natural bioactive compounds in therapeutic plants. Furthermore, when the purification of these phytochemicals is completed, then these molecules are tested for in vivo or in vitro anti-cancer effects. After achieving anti-cancerous results, other aspects such as pharmacokinetics, metabolic fate, side effects, immunogenicity, pharmacodynamics, dose determination, and drug interaction are focused on for future drug design [4].

## 4. Major Phytochemical Constituents of the Fabaceae Family

Species of the Fabaceae family are vital sources of phytochemicals, including flavonoids, lectins, saponins, alkaloids, carotenoids, and phenolic acids (Figure 5). These phytochemicals are present in every genus of the Fabaceae family and possess great medicinal values [18,22]. The phytochemicals have gained considerable recognition as far as anti-cancer properties are concerned [13]. However, the phytochemicals of the Fabaceae family have not been explored by various researchers around the world. According to available data, all these phytochemicals have significant anti-cancer values against different forms of cancers in humans (Figure 5). The structures of different phytochemicals found in different members of the Fabaceae family with anti-cancer values are presented in Figure 6.

Table 1 represents the data of different species of the family Fabaceae that exhibit anti-cancer activities. The major phytochemicals of the family Fabaceae, including alkaloids, flavonoids, carotenoids, lectins, phenolic acid, saponins, and terpenoids, are explored. In total, the anti-cancer activities of the phytochemicals of 71 species are documented in Table 1. The data related to the phytochemicals of the family Fabaceae are scarce as far as anti-cancer activity is concerned. Therefore, the best possible data related to the phytochemical activity of family Fabaceae members against cancer are presented in the table. The details about the importance of each phytochemical are given in the following sections.

### 4.1. Flavonoids

Flavonoids are considered to be effective anti-oxidants and are known to exhibit anti-angiogenic activity. Various studies have reported that flavonoids inhibit the metabolic activation of carcinogens and stop the further growth of abnormal cells, which may develop into cancerous cells [36]. Flavonoids and their derivatives are considered the vital phytochemical constituents of the Fabaceae family. The most important flavonoids isolated from the various members of the family are chalcone, flavones, flavonol, isoflavones, a flavonol glycoside, prenylated flavonoids, and lavandulyl flavanones [37]. According to Krishna et al. [38], prenylated flavonoids from various members of the Fabaceae family are known to exhibit anti-oxidant and anti-cancer activities.

Earlier, Kleemann et al. [39] reported that flavonoids could be used as a protectant against inflammation, cellular oxidation, and certain cancers. On the other hand, isoflavones extract from legume sprouts had inhibitory properties against breast cancer MCF-7 [13]. Wang et al. [40] also found that the isoflavones extract from *Cicer arietinum* L. have a repressive effect on MCF-7 breast cancer cells. Flow cytometry results and microscopic observations supported the inhibitory effect on MCF-7 cell lines, and *C. arietinum* isoflavones with a concentration of 32 µg mL^−1^ are enough to cause apoptosis of MCF-7.

Clinical studies have confirmed that there is a positive effect of isoflavones on human health by preventing various types of cancer, especially hormone-dependent cancers [41,42]. *Eriosema* (DC.) Desv. is another important genus in the Fabaceae family with anti-cancer activities. Flavonoids from species such as *Eriosema chenense* Vogel, *E. griseum* Baker, and *E. robustum* Baker have an inhibitory effect against various forms of cancer including lung cancer and oral epidermal carcinoma [43]. Aregueta-Robles et al. [44] found that *Phaseolus vulgaris* L. extract, as well as its flavonoid contents, have an inhibitory effect on lymphoma in mice both in vivo and in vitro. Flavonoid fraction stopped the production of cancerous cells in a dose-dependent way and, as a result, there was an increase in the cellular population at the S phase after the treatment with flavonoid fraction. Ombra et al. [45] also confirmed that flavonoids from *P. vulgaris* show considerable anti-cancer properties, and suppress the development of human MCF-7, while flavonoids also showed an inhibitory effect against human epithelial colorectal adenocarcinoma (Caco-3) cells. Moreover, Gatouillat et al. [46] reported the anti-cancer activity from the flavonoids fraction of genus *Medicago* L. and found that two flavonoids, namely, millepurpan and medicarpin isolated from *Medicago sativa* L., suppress cancer cells’ proliferation. According to Bora and Sharma [47], millepurpan and medicarpin can be utilized as chemopreventive agents for breast cancer as well as cervical cancer.

Stochmal et al. [48] investigated the role of flavone tricin as a chemopreventive agent sourced from *Medicago truncatula* Gaertn. It was noticed that tricin in humans caused cell cycle arrest or a growth inhibitory effect on MDA-MB-468 breast cancer. Tricin majorly inhibits the cyclooxygenase enzyme activity; therefore, it regulates the cyclooxygenase-mediated prostaglandin production. Due to this effect, tricin can be exploited as a chemopreventive agent for prostate and intestinal carcinogenesis. Custodio et al. [49] investigated another genus, *Ceratonia* L., from the Fabaceae family and reported that flavonoids extracted from *Ceratonia siliqua* L. have an inhibitory effect on tumor cell growth under in vitro conditions. Fu et al. [50] reported a novel flavonoid known as licochalcone-A from the roots of *Glycyrrhiza glabra* L., and this novel flavonoid leads to late G_1_ and G_2_ arrest in androgen-independent PC-3 prostate cancer cells. Choedon et al. [51] determined the effect of butrin extracted from the flowers of *Butea monosperma* (Lam.) Taub. on liver cancer and found significant results.

### 4.2. Lectins

Most members of the Fabaceae family are rich in lectin proteins, and communities in various regions use these plants for different diseases due to their anti-cancer and anti-tumor activities [13]. Several studies have confirmed the tumor inhibition mechanisms of lectins in various cell lines including bone, skin, bile duct, and liver cell lines [52,53,54,55,56]. According to De Mejia and Prisecaru [57], various forms of lectins showed anti-cancer properties under in vivo and in vitro conditions. Lectins bind with cancer cell membranes or their receptors resulting in apoptosis and cytotoxicity, and finally suppress the cancer cell growth. Fang et al. [58] assessed the anti-cancer activity of lectin isolated from the *Phaseolus vulgaris* L. and declared that lectin possesses anti-cancer activities, particularly against MCF-7, nasopharyngeal carcinoma cells (HNE-2, CNE-1, CNE-2), and liver cancer cells (Hep G2). Moreover, *P. vulgaris* lectin regulates nitric oxide (NO) production through the upregulation of inducible NO synthase known to introduce apoptotic bodies and contribute to the anti-carcinogenic activity. Similar results were confirmed by Lam and Ng [59] while working on the anti-cancer properties of lectin isolated from *P. vulgaris*.

Ye and Ng [60] demonstrated that lectin from *Glycine max* (L.) Merr. possesses anti-tumor properties for breast cancer and hepatoma cells. *C. arietinum* is a rich source of lectin and has a long history of medicinal use in several parts of India as it exhibits strong cancer chemopreventive activity [61]. Une et al. [62] purified the lectin from *Canavalia gladiata* (Jacq.) DC. using the DEAE-sephacel column and affinity chromatography and confirmed chemopreventive activity. Cavada et al. [63] found that *Conyza bonariensis* L. contains a considerable amount of lectin, which inhibits the process of carcinogenesis. According to Arteaga et al. [64], the proliferation of colon cancer cells is greatly inhibited by the lectin isolated from *Phaseolus acutifolius* A. Gray. Gondim et al. [65] evaluated the anti-cancer activity of seed isolated DLasiL lectin of *Dioclea lasiocarpa* Mart. Ex Benth. The experiments showed that DLasiL lectin was effective against PC-3 prostate cancer, A-2780 ovarian, and MCF-7 breast cancer cell lines.

Lagarda-Diaz et al. [66] stated that legume lectins showed anti-oxidant and anti-cancer activities. Legume lectins inhibit cell proliferation in lung cancer, and the consumption of legume lectins contributes to immunity against different forms of cancer. Korourian et al. [67] used *Griffonia simplicifolia* (DC.) Baill. lectin-1 (GS 1) to suppress the progression of human breast ductal carcinoma and found significant results.

**Table 1 molecules-27-03863-t001:** Phytochemicals of family Fabaceae and suppression of particular cancer types.

S. No.	Species	Genus	Phytochemicals	Targeted Cancer	References
1	*Acacia nilotica* (L.) Willd. eX Del.	*Acacia*	Gallic acid	Not specified	[68]
2	*Acacia hydaspica* R. parker	*Acacia*	Alkaloids, flavonoids, and saponin	Not specified	[69]
3	*Acacia saligna* (Labill.) H.L.Wendl.	*Acacia*	Flavonoids and saponin	Hep G2 cancer (liver cancer)	[70,71]
4	*Acacia seyal* Delile	*Acacia*	Lectin	Hepatocellular carcinoma, HEP-2 (Larynx Cancer), HCT116 (colon cancer), and MCF-7 (breast cancer)	[72]
5	*Acacia victoriae* Benth.	*Acacia*	Avicins and Fo35	Breast cancer	[73]
6	*Albizia lebbeck* (L.) Benth.	*Albizia*	Saponins, flavonoids	Liver, larynx, breast, cervical, and colon cancer	[74,75]
7	*Albizia chinensis* (Osbeck) Merr.	*Albizia*	Quercetin (Flavonoid)	Myeloid leukemia	[76,77]
8	*Albizia Julibrissin* Baker	*Albizia*	Alkaloids, saponins, and flavonoids	Leukemia	[78,79,80]
9	*Astragalus ovinus* Boiss.	*Astragalus*	Phenolics, flavonoids	Breast cancer in rats	[81]
10	*Astragalus spinosus* (Forssk.) Muschl.	*Astragalus*	Flavonoids	Not specified	[82]
11	*Astragalus membranaceus* (Fisch.) Bunge	*Astragalus*	Flavonoids, saponins	Breast cancer	[83]
12	*Bauhinia acuminata* L.	*Bauhinia*	Alkaloids, flavonoids	Lung cancer	[84]
13	*Bauhinia variegata* (L.) Benth.	*Bauhinia*	Alkaloids, Kaempferol galactoside, saponins	Liver, lung, breast cancer (both in vitro and in vivo), and human ovarian cancer (in vivo)	[85,86,87]
14	*Bauhinia purpurea* L.	*Bauhinia*	Lectin	MCF-7 (breast cancer)	[88]
15	*Butea monosperma* (Lam.) Taub.	*Butea*	Butrin	Liver cancer	[51]
16	*Caesalpinia bonduc* (L.) Roxb	*Caesalpinia*	Alkaloids	Not specified	[89]
17	*Caesalpinia gilliesii* (Hook.) D.Dietr	*Caesalpinia*	Isorhamnetin, Isorhamnetin-3-O-rhamnoside (flavonoids)	MCF-7 (breast cancer) and HepG2 cancer (liver cancer)	[90]
18	*Caesalpinia pluviosa* DC.	*Caesalpinia*	Caesalpinioflavone	A549 (lung adenocarcinoma), MCF-7, and Hst578T (breast cancer)	[91]
19	*Caesalpinia pulcherrima* (L.) Sw.	*Caesalpinia*	Catechin, Gallic acid, quercetin, Rutin	Breast cancer	[92]
20	*Cajanus cajan* (L.) Millsp.	*Cajanus*	Flavanones	CaCo-2 (colorectal) HeLa (cervical), and MCF-7 (breast cancer) cancer	[93,94,95]
21	*Canavalia gladiata* (Jacq.) DC.	*Canavalia*	Lectin	Not specified	[62]
22	*Cassia occidentalis* (L.) Link	*Cassia*	Alkaloids, flavonoids, saponins	HCT-15, SW-620 (colon cancer), OVCAR-5 (ovarian cancer), SiHa (cervical cancer), PC-3 (prostate cancer, and MCF-7 (breast cancer)	[96,97]
23	*Castanospermum australe* A.Cunn. eX mudie	*Castanospermum*	Castanospermine	Not specified	[98]
24	*Ceratonia siliqua* L. (carob)	*Ceratonia*	Flavonoids	Not specified	[99]
25	*Cicer arietinum* L.	*Cicer*	Isoflavones	Breast cancer	[40]
26	*Conyza bonariensis* L.	*Conyza*	Lectin	Not specified	[63]
27	*Cytisus villosus* Pourr.	*Cytisus*	Flavonols, flavones	Breast and colon cancer	[100]
28	Derris scandens Roxb. (Benth.)	*Derris*	Glyurallin, derrubone, derriscandenon B and C (isoflavones)	HT29 (colon cancer)	[95,101]
29	*Dioclea lasiocarpa* Mart. eX Benth.	*Dioclea*	Lectin DLasiL	Breast, prostate, and ovarian cancer	[65]
30	*Eriosema chinense* Vogel	*Eriosema*	Isoflavone, flavonols	Lung cancer and oral epidermal carcinoma	[43]
31	*Eriosema griseum* Baker	*Eriosema*	Flavonols, flavanones	Lung cancer and oral epidermal carcinoma	[43]
32	Erythrina senegalensis DC	*Erythrina*	Alkaloids, flavonoids	Breast, cervical, colon, liver, lung cancer, and leukemia	[102]
33	*Gleditsia triacanthos* L.	*Gleditsia*	Flavones	Liver, breast, cervical, larynx, and colon cancer	[103]
34	*Gleditsia caspica* Desf.	*Gleditsia*	Saponins	MCF-7 (breast cancer)	[77]
35	*Gleditsia sinensis* Lam.	*Gleditsia*	Saponins	MCF-7 (breast cancer)	[104]
36	*Glycine max* (L.) Merr.	*Glycine*	Lectin, genistein (Isoflavones), saponins	Breast and liver cancer	[60,67]
37	*Glycyrrhiza uralensis* Fisch. eX DC.	*Glycyrrhiza*	Isoliquiritigenin	Human lung cancer (in vitro)	[105]
38	*Glycyrrhiza glabra* L.	*Glycyrrhiza*	Alkaloids, flavonoids, saponins	Breast, colon, liver, and prostate cancer	[50,106]
39	*Griffonia simplicifolia* (DC.) Baill.	*Griffonia*	Lectin-1	Breast cancer	[67]
40	*Indigofera tinctoria* L.	*Indigofera*	Flavonoids, phenolic compounds, Saponins	Lung cancer	[107,108,109]
41	*Indigofera cassioides* Rottl. Ex. Dc.	*Indigofera*	Flavonoids, saponins, terpenoids	Breast and colon cancer (in vitro and in vivo)	[110]
42	*Indigofera aspalathoides* (Vahl.)	*Indigofera*	Alkaloids, flavonoids, saponins	Cervical cancer	[111]
43	*Indigofera cordifolia* B.Heyne eX Roth	*Indigofera*	Alkaloids, flavonoids, saponins	Human breast, cervical, liver, and lung cancer	[112]
44	*Indigofera suffruticosa* Mill.	*Indigofera*	Alkaloids, flavonoids, lectin	Not specified	[113]
45	*Laburnum anagyroides* Medik.	*Laburnum*	Cytisine	Lung cancer	[114]
46	*Medicago arabica* (L.) Huds.	*Medicago*	Saponins	HeLa (cervical cancer)	[115]
47	*Medicago Sativa* L.	*Medicago*	Alkaloids, millepurpan, medicarpin (flavonoids), saponins	Breast and cervical cancer	[46,116,117,118]
48	*Medicago truncatula Gaertn.*	*Medicago*	Tricin (flavone)	Breast cancer, intestinal carcinogenesis and prostate cancer	[48]
49	*Melilotus officinalis* (Linn.) Pall.	*Melilotus*	Saponins	Prostate cancer	[119]
50	*Melilotus indicus* (L.) All.	*Melilotus*	Flavonoids	Hepatocellular carcinoma	[120]
51	*Parkia javanica* Lam.	*Parkia*	Alkaloids, flavonoids, saponins	Human liver cancer	[121]
52	*Phaseolus vulgaris* L.	*Phaseolus*	Galic acid, lectin	Breast cancer, colon cancer, Epithelial colorectal adenocarcinoma, liver cancer, and nasopharyngeal carcinoma	[45,58]
53	*Phaseolus Acutifolius* A. Gray	*Phaseolus*	Lectin	Colon cancer	[64]
54	*Physostigma venenosum* Balf.	*Physostigma*	Physostigmine alkaloid or eserine	Not specified	[122]
55	*Prosopis juliflora* (Sw.) DC.	*Prosopis*	Alkaloids	Leukemia	[123,124]
56	*Prosopis cineraria* (L.) Druce	*Prosopis*	Alkaloids, flavonoids, phenolic aicd, saponins	Hepatocellular carcinoma	[125,126]
57	*Pseudarthria hookeri* Wight & Arn.	*Pseudarthria*	Flavanones, flavones, isoflavone	Epithelial colorectal adenocarcinoma (CaCo-2), Leukemia, lung adenocarcinoma (A549), and human ovarian carcinoma (Skov-2)	[127]
58	*Psoralea corylifolia* L.	*Psoralea*	Neobavaisoflavone (flavonoids)	Colon cancer and leukemia	[128]
59	*Senna alexandrina* Mill.	*Senna*	Flavonoids	Liver cancer	[129]
60	*Sesbania grandiflora* (L.) poiret	*Sesbania*	Alkaloids, flavonoids, and saponins	Colon cancer	[130,131]
61	*Sophora tonkinensis* Gagnep.	*Sophora*	Isoflavones	Breast cancer	[132]
62	*Sophora flavescens* Aiton	*Sophora*	Oxymatrine (Alkaloid)	Cervical, colorectal, gastric, human hepatoma carcinoma, lung, pancreatic, and laryngeal cancer	[76,133,134,135,136,137,138,139,140,141]
63	*Spatholobus suberectus* Dunn	*Spatholobus*	Flavonoids, phenolic acid	Not specified	[142]
64	*Tephrosia purpurea* L.	*Tephrosia*	Flavonoids	MCF-7 (breast cancer)	[143]
65	*Trifolium repens* L.	*Trifolium*	Flavonoids, alkaloids	Not specified	[144]
66	*Trifolium spinosa* L.	*Trifolium*	Flavonoids, alkaloids	Not specified	[144]
67	*Trifolium pretense* L.	*Trifolium*	Flavonoids	Breast cancer	[145]
68	*Trigonella foenum-graecum* L.	*Trigonella*	Apigenin, luteolin (flavone)	Breast, colon, esophageal squamous cell carcinoma, lung, and prostate cancer	[146,147]
69	*Vicia faba* L.	*Vicia*	Flavonoids	MCF-7 (breast cancer), HCT 116	[23]
70	*Wisteria sinensis* (Sims) DC.	*Wisteria*	Flavonoids	Hepatocellular Carcinoma	[148]
71	*Wisteria floribunda* (Willd.) DC.	*Wisteria*	Lectin	MCF-7 (breast cancer)	[88]

### 4.3. Saponins

Several members of the Fabaceae family including peanut, soybean, and lentil are rich in saponins and reported to exhibit anti-cancer properties. Various researchers around the globe have confirmed that saponins isolated from members of the Fabaceae family are effective against colon cancer, melanoma cells, and cervical cancer. Saponins can follow various mechanisms to suppress the progression of cancer by cell cycle arrest, the inhibition of cellular invasion, anti-oxidant activity, and the induction of autophagy and apoptosis [149]. Rochfort and Panozzo [150] stated that the intake of legume saponins enhances the immunity against various types of cancer including cervical and colon cancer. Mudryj et al. [151] examined the anti-carcinogenic activity of legume saponins and reported that saponins involve different mechanisms such as immune modulatory effects, acid and neutral sterol metabolism, the normalization of carcinogen-induced cell proliferation, and cytotoxicity of cancerous cells.

Saponins show growth-repressing effects against colon cancer cells by interacting with cholesterol or free sterols that occur in the cell membranes and lead to a change in its permeability [152]. According to Gurfinkel and Rao [153], microorganisms in the colon hydrolyze saponins to sapogenols, which act as a strong chemopreventive agent against colon cancer and delimit further cancer progression. Dai et al. [154] stated that the intake of saponins from *Glycine max* reduces the risk of and controls breast cancer growth. The effects were more significant, in particular, in the case of premenopausal women. Furthermore, saponins from *Glycine max* also inhibit prostate cancer; however, more efforts from researchers are required to understand the exact mechanism. Mujoo et al. [73] demonstrated that saponins are present in substantial amounts in *Acacia victoriae* Benth., which inhibit the proliferation of various tumor cell lines with minimal growth inhibition in immortalized breast epithelial cells, human foreskin fibroblasts, and mouse fibroblasts at a similar concentration. Mujoo et al. [73] also investigated two saponins (avicins and Fo35) from *A. victoriae* and reported that both cause apoptosis of the Jurket (T-cell leukemia), cell cycles arrest (G1) of the human breast cancer cell line (MDA-MB-453), and apoptosis of cancer cell line (MDA-MB-435).

### 4.4. Alkaloids

Alkaloids are vital secondary metabolites that are considered a valuable source of novel drugs. Several studies have confirmed that alkaloids have anti-cancer and anti-proliferative properties [155]. Vindesine, vinorelbine, vinblastine, and vincristine are the best examples of alkaloids, which have already been successfully developed as anti-cancer drugs. These are effective against different forms of cancer including testicular cancer, brain cancer, lung cancer, bladder cancer, and melanoma. Over 21,000 different alkaloids have been identified and most of these alkaloids are a great source of medicines, especially exhibiting anti-cancer activities [156].

Steroidal alkaloids are the most promising component of phytochemicals as far as the anti-cancer potential is concerned. Steroidal alkaloids could be used in the discovery of safer drugs for cancer treatment with the aid of more clinical experiments in the future [157]. Matrine alkaloid found in the members of the genus *Sophora* showed potential anti-cancer effects against lung cancer and liver cancer [158]. Oxymatrine is one of the few important quinolizidine alkaloid compounds extracted majorly from the roots of *Sophora flavescens* Aiton. Oxymatrine is reported to increase the anti-tumor immunity against lung cancer and can be used to enhance the immunity against various other types of cancer [141]. Cytisine is another alkaloid naturally occurring in two genera of the Fabaceae family including *Cytisus* and *Laburnum* [159]. Cytisine is helpful in the suppression of lung cancer through the induction of mitochondria-mediated apoptosis and cell cycle arrest and suggests potential anti-cancer activity [114].

Castanospermine is another alkaloid extracted from *Castanospermum australe* A. Cunn ex Hook. [98] and is reported to convert protein N-linked high mannose carbohydrates to complex oligosaccharides. Castanospermine serves as an inhibitor of the glycosidases and leads to the suppression of tumor cell proliferation in nude mice [160]. Physostigmine alkaloid, also known as eserine, occurs naturally in *Physostigma venenosum* Balf. and exhibits anti-tumor activities [122]. Pfitzinger et al. [122] reported that physostigmine treatment significantly suppresses tumor-associated inflammation in mice. However, alkaloids in the Fabaceae family have not been explored in the same way as other family members as far as anti-cancer activities are concerned; therefore, more research is required for the further discovery of potent anti-cancer drugs from alkaloids that occur in the Fabaceae family.

### 4.5. Carotenoids

Legume leaves are a vital source of carotenoids, which primarily include carotenes, while other carotenoids are lutein, neoxanthin, crocetin, antheraxanthin, violaxanthin, and some others in a very low quantity. On the other hand, legume roots are not as rich in carotenoids as the leaves [161]. Many experimental studies have identified various mechanisms through which carotenoids may control the development of various types of cancer in humans. These mechanisms include anti-oxidant actions, retinol, communication functions, and cell signaling. Therefore, anti-oxidant defense support from the carotenoids reduces cancer risks [162]. Nishino et al. [163] carried out an extensive study and reported that β-carotene, β-cryptoxanthin, lycopene, lutein, and zeaxanthin can be used as chemopreventative agents. Moreover, beta-cryptoxanthin regulates the expression of the RB gene, which is a known anti-oncogene. Horvath et al. [164] stated that the carotenoids extracted from legumes have protective, preventative, and even curative effects against various types of cancer.

Lutein and zeaxanthin are two vital carotenoids that lower the risk of certain cancers [165]. Cancer is associated with the inflammation processes; therefore, the beneficial effects of both lutein and zeaxanthin are due to anti-inflammatory and anti-oxidant properties [164,166], but the exact mechanism of lutein and zeaxanthin action is not clearly understood and needs to be explored in future studies. Beta-carotene markedly inhibits the growth of esophageal cancerous cells in a time- and dose-dependent manner. Another significant fact is that the same concentration of beta-carotene is non-toxic to normal esophageal epithelium Het-lA cells, suggesting β-carotene is a potent anti-cancer agent [167]. PC-SPES is a patented herbal mixture that is utilized in prostate cancer treatment, and this herbal mixture is unique in its composition as it is a combination of eight herbs, two of which belong to the Fabaceae family, including *Glycyrrhiza glabra* L. and *G. uralensis* Fisch. ex DC [168]. According to Matus et al. [169], terpenoids are significant inhibitors of the signaling of NF-kB, which is a key regulator in cancer and inflammation. Carotenoids can use a variety of pathways for their anti-cancer activity; however, the induction of apoptosis is considered the most common.

Satia et al. [170] reported that the long-term use of retinol, β-carotene, lutein, and lycopene reduces the risk of lung cancer. Gong et al. [171] stated that legumes are an important source of lutein and significantly inhibit the proliferation of breast cancer cells and enhance the effect of chemopreventive agents through reactive oxygen species (ROS)-mediated mechanisms. Rafi et al. [172] determined the effect of lutein on the proliferation of human prostate cancer cells (PC3) as well as rat prostate carcinoma cells (AT3 cells). The anti-cancer activity of lutein was effective against both rat and human prostate cancer. Kim et al. [173] demonstrated that zeaxanthin in combination with lutein lowers the risk of colorectal cancer through apoptosis of cancerous cells and anti-oxidant functions.

### 4.6. Phenolic Acids

Phenolic acids are vital phytochemicals present in considerable amounts in the members of the Fabaceae family. Phenolic acids are non-flavonoid phenolic compounds that occur in the free, insoluble-bound, and conjugated soluble forms. On the other hand, these non-flavonoid phenolic compounds are widely distributed in plant species [174]. Natural phenolic acids present in various members of the Fabaceae family are ferulic acid, vanillic acid, caffeic acid, benzoic acid, p-hydroxy acid, 3,4-dihydroxybenzoic acid, sinapinic acid, and syringic acid [175]. Phenolic acids are secondary compounds that have been explored recently against various diseases, particularly cancer. These phenolics reduce the proliferation of cancerous cells, promote apoptosis, and target various aspects of cancer including growth, development, and metastasis [176]. Recently, phenolic acids have been extensively studied due to their anti-inflammatory, anti-tumor, and anti-oxidant activities [177]. Anantharaju et al. [178] demonstrated that the anti-carcinogenic effect of phenolic acids is largely due to five activities: (1) modulation of ROS levels, (2) inducing cell cycle arrest, (3) promoting the suppression of tumor proteins such as p53, (4) suppressing oncogenic signaling cascades controlling apoptosis and angiogenesis as well as proliferation, (5) increasing the ability to differentiate and, finally, transforming into normal cells.

Palko-Labuz et al. [179] stated that phenolic acids exhibit numerous health-related benefits such as anti-oxidant, anti-cancer, and anti-inflammatory activities. Phenolic acids have low bioavailability, which often restricts their possible medical applications; however, conjugation with phospholipids could be helpful to enhance the bioavailability in the biological system. The results showed that conjugates were effective as apoptosis-inducing, anti-proliferative, and cell cycle-affecting agents. Moreover, the same concentration was effective for the majority of metastatic melanoma cell lines and, importantly, did not affect the normal fibroblasts. Salem et al. [68] isolated bioactive gallic acid from the pod extract of *Acacia nilotica* (L.) Willd. ex Dilile and reported that gallic acid has anti-tumor properties due to anti-oxidant and anti-inflammatory properties.

## 5. Conclusions and Future Directions

Species of the Fabaceae family are a rich source of phytochemicals including flavonoids, lectins, saponins, alkaloids, carotenoids, and phenolic acids. The consumption of various species of the Fabaceae family lowers the risk of cancer, as the phytochemicals from Fabaceae members are effective in the prevention and treatment of cancer. Some of these phytochemicals have already been utilized against cancer worldwide; however, other phytochemicals are also gaining importance. These phytochemicals use a variety of mechanisms to control cancer including carcinogen inactivation, the induction of cell cycle arrest, anti-oxidant stress, apoptosis, and regulation of the immune system. On the other hand, there is room for more research to be carried out to assess the anti-cancer properties of phytochemicals of the Fabaceae family. More data are needed relating to the phytochemicals of the Fabaceae family for anti-cancer properties, and these data would lead to the discovery of novel drugs from these phytochemicals. Similarly, more studies elucidating the mechanisms behind the anti-cancer properties of phytochemicals are also required in the future. Despite the effectiveness of different phytochemicals in a plant belonging to the Fabaceae family, there is a need to elucidate any synergistic impact of different anti-cancer phytochemicals in a single plant. In the future, any long-term adverse side effects in terms of physiological changes in patients caused by different anti-cancer phytochemicals found in the Fabaceae family also require elucidation. Despite many reports about the efficacy of different anti-cancer phytochemicals, most of these reports are under in vitro or in vivo experiment conditions, and very few clinical trial reports are available. Therefore, more clinical trial reports confirming the efficacy of phytochemicals from Fabaceae members with responsible mechanisms will be indispensable in future studies. To achieve the international standard, significant standardization of prospective phytochemicals in terms of techniques for analyzing their bioavailability, efficacy, safety, quality, composition, manufacturing processes, and regulatory and approval requirements are required.

## Figures and Tables

**Figure 2 molecules-27-03863-f002:**
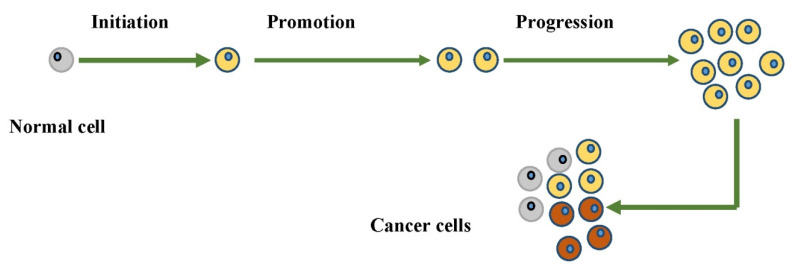
Diagrammatic representation of carcinogenesis process.

**Figure 3 molecules-27-03863-f003:**
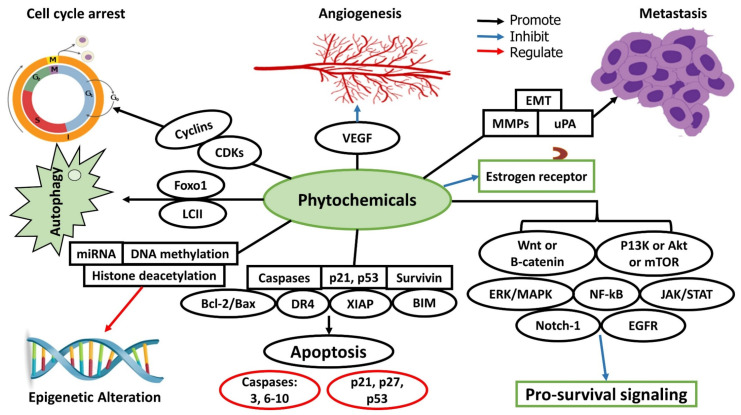
Phytochemicals’ pathway of anti-cancer effect.

**Figure 4 molecules-27-03863-f004:**
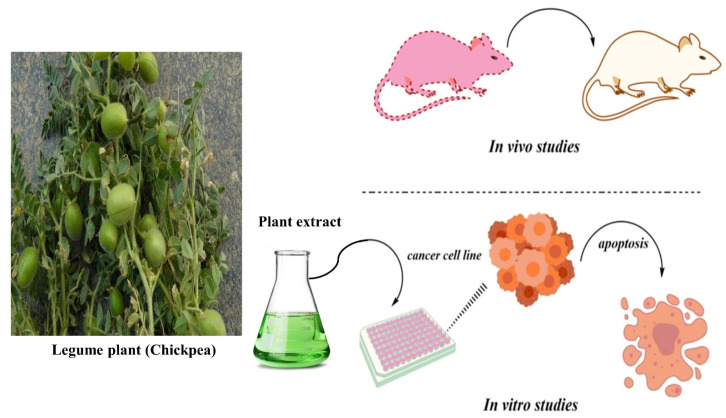
Assessment of phytochemicals from medicinal plants for anti-cancer activity.

**Figure 5 molecules-27-03863-f005:**
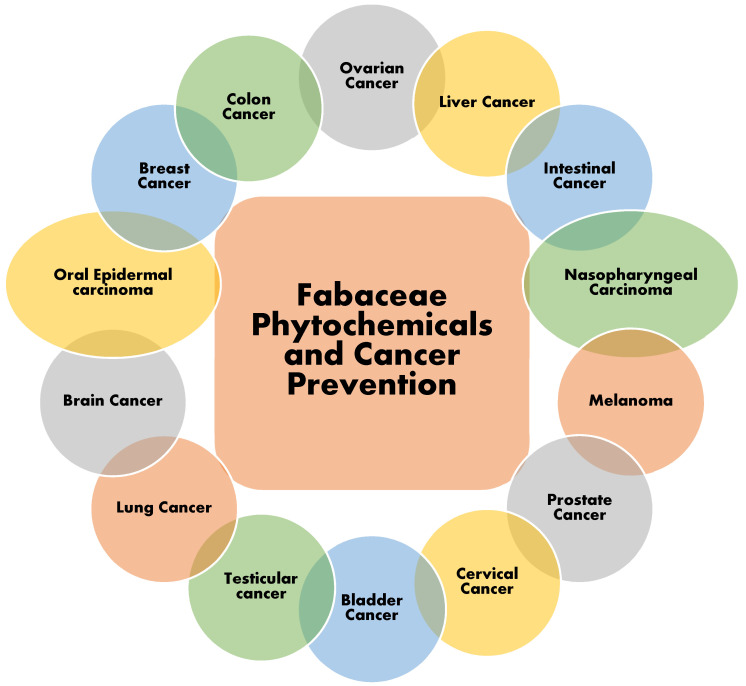
Reported activity of phytochemicals of family Fabaceae against various types of cancers.

**Figure 6 molecules-27-03863-f006:**
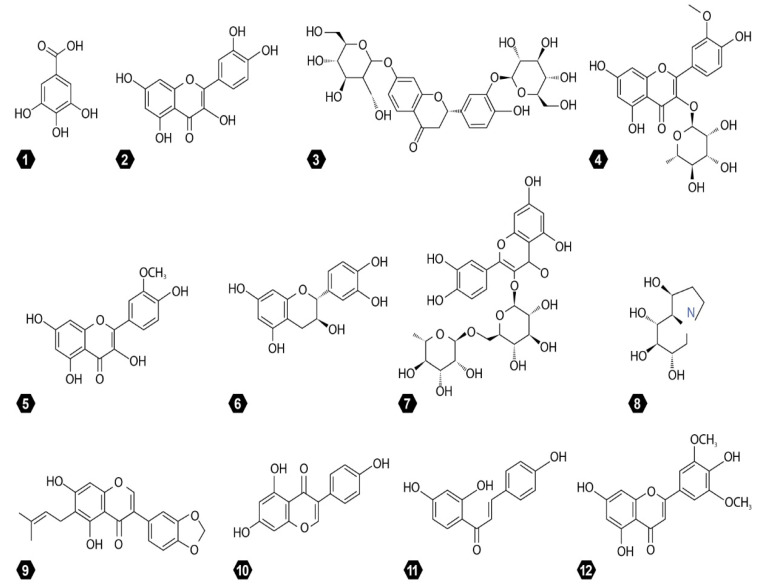
Structure of main phytochemicals from family Fabaceae. 1. Gallic acid. 2. Quercetin. 3. Butrin. 4. Isorhamnetin-3-O-rhamnoside. 5. Isorhamnetin. 6. Catechin. 7. Rutin. 8. Castanospermine. 9. Derrubone. 10. Genistein. 11. Isoliquiritigenin. 12. Tricin.

## Data Availability

Not applicable.

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
