# Peer review of "Exploring the Phytochemicals and Anti-Cancer Potential of the Members of Fabaceae Family: A Comprehensive Review"

_molecules, 2022, doi:10.3390/molecules27123863_

Round 1

Reviewer 1 Report

THE PRESENT REVIEW EXPLAIN THE ANTICANCER POTENTIAL OF PHYTOCHEMICALS FROM THE PLANTS BELONGS TO FABACEAE FAMILY. THE OBJECTIVE, LIMITATIONS AND FUTURE DIRECTIONS ARE WELL EXPLAINED. IT WOULD BE APPRECIATED IF THE GRAPHICAL REPRESENTATION IS GIVEN FOR CANCER STATISTICS INSTEAD OF TABLE. THE PHYTOCHEMICALS AND THEIR PATHWAY OF ANTICANCER EFFECT ALSO CAN BE GIVEN IN THE FORM OF IMAGES. THIS WILL GIVE MORE UNDERSTANDING ABOUT THE MOLECULAR MECHANISMS. THE IMAGES WHICH IS GIVEN IN THE MANUSCRIPT IS NOT INFORMATIVE. IT WILL BE GOOD IF THE AUTHORS GIVE MORE DETAILED IMAGES (FIG. 1-3). IN TABLE 2, S. No. 41, PLEASE CHECK THE NAME OF THE PLANT ( Indigofera cassioides Rottl. Ex. DC). S. No. 58, IT IS COLON CANCER. IF POSSIBLE THE AUTHORS CAN GIVE THE STRUCTURES OF THE PHYTOCHEMICALS DISCUSSED IN THE REVIEW. THE WORK IS CONCLUDED WELL.

Author Response

Response to reviewer 1 comments

Comment: The present review explain the anticancer potential of phytochemicals from the plants belongs to Fabaceae family. The objective, limitations and future directions are well explained.

Response: Thanks for your appreciation

Comment: It would be appreciated if the graphical representation is given for cancer statistics instead of table.

Response: We have represented Table data in graphical representation as Figure 1 and in more concise form as per the suggestion of the reviewer

Comment: The phytochemicals and their pathway of anticancer effect also can be given in the form of images. This will give more understanding about the molecular mechanisms. The images which is given in the manuscript is not informative. It will be good if the authors give more detailed images (fig. 1-3).

Response: The phytochemicals and their pathway of anticancer effect are now given in the figure 3

Comment: In table 2, s. No. 41, please check the name of the plant (indigofera cassioides rottl. Ex. Dc). S. No. 58, it is colon cancer. If possible the authors can give the structures of the phytochemicals discussed in the review.

Response: Both checked, corrected and highlighted

Comment: The work is concluded well.

Response: Thanks for your appreciation

Reviewer 2 Report

M. Usman et al. report herein an extensive description of Exploring the Phytochemicals and Anti-Cancer Potential of the Members of Fabaceae Family. The authors present a complete overview of recent publications in this field. 

The manuscript is well written and organized, with appropiate references, and covers a hot topic in current Natural Products Chemistry research. 

For all these reasons, I encourage publication in Molecules in its present form.

Author Response

Response to the reviewer 2 comments

Comment: M. Usman et al. report herein an extensive description of Exploring the Phytochemicals and Anti-Cancer Potential of the Members of Fabaceae Family. The authors present a complete overview of recent publications in this field. The manuscript is well written and organized, with appropriate references, and covers a hot topic in current Natural Products Chemistry research. For all these reasons, I encourage publication in Molecules in its present form.

Response: Thanks for your appreciation and recommendation

Reviewer 3 Report

This paper explores the anti-cancer potential of the members of Fabaceae family by summarizing the main chemical components, including flavonoids, lectins, saponins, alkaloids, carotenoids, phenolic acids, and the research on the anticancer activity. The topic is interesting; however, this paper seems to lack the authors' thoughts and comments and is more like mere 179 documents arrangement. A large number of the length in this article is devoted to plant chemical components, but less to the anti-cancer prospect. The specific opinions are as follows.

1.        It is suggested to remove "drug discovery" from the "Keywords". Separate keywords with “,” instead of “;”.

2.        The abstract part needs to be further simplified and improved. Line 28-29: “There is huge pressure on the pharmaceutical industry due to the increased demand for anti-cancer drugs” is not needed. The importance of the use of phytochemicals in the Fabaceae family needs to be emphasize.

3.        Line 44: “Many stages occur in the formation of the cancerous cells” The stages here can be listed briefly rather than just one sentence.

4.        Line 95: The subtitle here does not summarize what is said in the next two paragraphs.

5.        The content in Table 1 does not have a clear relationship with the main purpose of the article and is recommended to be concise or deleted.

6.        The introduction can be more concise and clearer.

7.        The layout of figures can be optimized.

8.        It is better to replace the figure of “Diagrammatic representation of carcinogenesis process.” with a vivid and exquisite one.

9.        The manuscript lacks in-depth discussion and prospects. The discussion and prospect part did not indicate their own ideas, did not put forward effective and referential research directions or paths, and repeated the contents of the abstract.

10.    Pay attention to the reference order in the paper. Number them from lowest to highest in the order in which they appear in the paper.

11.    The wording is repetitive and lengthy, and the language lacks refinement. Many grammar issues need to be fixed.

Author Response

Response to the reviewer 3 comments

Comment: This paper explores the anti-cancer potential of the members of Fabaceae family by summarizing the main chemical components, including flavonoids, lectins, saponins, alkaloids, carotenoids, phenolic acids, and the research on the anticancer activity. The topic is interesting; however, this paper seems to lack the authors' thoughts and comments and is more like mere 179 documents arrangement. A large number of the length in this article is devoted to plant chemical components, but less to the anti-cancer prospect. The specific opinions are as follows.

Response:

Comment 1. It is suggested to remove "drug discovery" from the "Keywords". Separate keywords with “,” instead of “;”.

Response: We have removed “drug discovery” from keywords and added “antioxidants; apoptosis”. Regarding the separation of keywords with “,” instead of “;”, we have cross-checked the authors' instructions and template, the keywords as per the format of the paper

Comment 2. The abstract part needs to be further simplified and improved. Line 28-29: “There is huge pressure on the pharmaceutical industry due to the increased demand for anti-cancer drugs” is not needed. The importance of the use of phytochemicals in the Fabaceae family needs to be emphasize.

Response: Line 28-29 is removed whereas the importance of the use of phytochemicals is emphasized as well.

Comment 3. Line 44: “Many stages occur in the formation of the cancerous cells” The stages here can be listed briefly rather than just one sentence.

Response: We have added different stages as “initiation, promotion, and progression” in the mentioned sentence

Comment 4. Line 95: The subtitle here does not summarize what is said in the next two paragraphs.

Response: We have modified the subtitle as per the suggestion of the reviewer

Comment 5. The content in Table 1 does not have a clear relationship with the main purpose of the article and is recommended to be concise or deleted.

Response: We have made the data in Table 1 concise and presented it in Figure 1 

Comment 6. The introduction can be more concise and clearer.

Response: We have removed the unnecessary information to make the introduction more concise and clearer.

Comment 7. The layout of the figures can be optimized.

Response: We have modified the figure as per the suggestion of the reviewer

Comment 8. It is better to replace the figure of “Diagrammatic representation of carcinogenesis process.” with a vivid and exquisite one.

Response: We have replaced the figure of “Diagrammatic representation of carcinogenesis process.” with a vivid and exquisite one as per the suggestion of the reviewer

Comment 9. The manuscript lacks in-depth discussion and prospects. The discussion and prospect part did not indicate their own ideas, did not put forward effective and referential research directions or paths, and repeated the contents of the abstract.

Response: We have modified the mentioned portions and put forward effective and referential research directions, removed the repeated information

Comment 10. Pay attention to the reference order in the paper. Number them from lowest to highest in the order in which they appear in the paper.

Response: We have cross-checked and arranged the number of the reference from lowest to highest in the order in which they appear in the paper

Comment 11. The wording is repetitive and lengthy, and the language lacks refinement. Many grammar issues need to be fixed.

Response: We have cross-checked, corrected, and proofread our manuscript from my Colleague Dr. Patrick Finnegan at UWA

Round 2

Reviewer 3 Report

This review summarizes the detailed research progress related to the phytochemical status of the Fabaceae family and their anti-cancer properties. The layout, pictures, and grammar of the revised paper are better. The paper can provide valuable research ideas for later researchers. I think there are some details in the paper that can be modified to make the article more specific.

1. If Figure 6 is replaced by the structure of the main anti-cancer phytochemicals of Fabaceae Family and adds discussion on structure, it could improve the value and significance of the paper.

2. Chapter 3 describes the steps of developing phytochemical drugs from medicinal plants. It is not strongly related to the theme of the article "legume chemical components" and "anti-cancer". It can specifically discuss the experimental techniques used in the anti-cancer research of legumes.